# *Megasphaera elsdenii:* Its Role in Ruminant Nutrition and Its Potential Industrial Application for Organic Acid Biosynthesis

**DOI:** 10.3390/microorganisms12010219

**Published:** 2024-01-21

**Authors:** Luciano da Silva Cabral, Paul J. Weimer

**Affiliations:** 1Department of Animal Science and Rural Extension, Agronomy and Animal Science School, Federal University of Mato Grosso, Cuiabá 780600-900, Mato Grosso, Brazil; luciano.cabral@ufmt.br; 2Department of Bacteriology, University of Wisconsin, Madison, WI 53706, USA

**Keywords:** carboxylate platform, lactic acid, *Megasphaera*, rumen, ruminants, volatile fatty acids

## Abstract

The Gram-negative, strictly anaerobic bacterium *Megasphaera elsdenii* was first isolated from the rumen in 1953 and is common in the mammalian gastrointestinal tract. Its ability to use either lactate or glucose as its major energy sources for growth has been well documented, although it can also ferment amino acids into ammonia and branched-chain fatty acids, which are growth factors for other bacteria. The ruminal abundance of *M. elsdenii* usually increases in animals fed grain-based diets due to its ability to use lactate (the product of rapid ruminal sugar fermentation), especially at a low ruminal pH (<5.5). *M. elsdenii* has been proposed as a potential dietary probiotic to prevent ruminal acidosis in feedlot cattle and high-producing dairy cows. However, this bacterium has also been associated with milk fat depression (MFD) in dairy cows, although proving a causative role has remained elusive. This review summarizes the unique physiology of this intriguing bacterium and its functional role in the ruminal community as well as its role in the health and productivity of the host animal. In addition to its effects in the rumen, the ability of *M. elsdenii* to produce C_2_–C_7_ carboxylic acids—potential precursors for industrial fuel and chemical production—is examined.

## 1. Introduction

Ruminant animals, including cattle, sheep, and goats, are characterized by a digestive tract in which the foregut is compartmentalized into four chambers. The first chamber, the rumen, accounts for around 65% of foregut volume and contributes to more than 65% of the digestion that occurs in the entire digestive tract of adult animals [1] (p. 230). Digestion that takes place in the rumen is exclusively attributed to its microbial community, which contains both a high density and high diversity of microbes, estimated at up 10^9^ to 10^11^ bacterial cells mL^−1^, 10^4^ to 10^6^ protozoa mL^−1^ [2] (p. 34), and 10^2^ to 10^4^ fungal zoospores mL^−1^ [3] of ruminal contents. Of particular importance is the digestion of polysaccharides within the plant cell wall, which cannot be digested by enzymes secreted by the animal. Ruminal digestion of feeds results in the production of microbial cells and volatile fatty acids (VFAs), which are, respectively, the major source of amino acids and energy for the host [4,5]. Since Hungate began to study the ruminal microbial ecosystem in the 1940s, around 450 bacterial species have been isolated from the rumen [6], but the total number of species is estimated to be >4000; therefore, the majority of the rumen’s bacterial diversity remains uncharacterized.

One ruminal bacterium, *Megasphaera elsdenii,* has been studied in particular for its capacity to ferment lactate, which is produced by various sugar-fermenting bacteria in cattle and sheep fed high-grain diets [7], thus making this species potentially important in controlling ruminal acidosis. This in turn has created expectations regarding its potential use as a probiotic. In addition to its importance in the rumen’s microbial ecology, *M. elsdenii* has also attracted attention for its ability to produce commercially useful short- and medium-chain carboxylic acids (“volatile fatty acids”, VFAs) [8]. Despite its long history of study, its unique physiology, and its potential as both a probiotic in ruminant livestock and an industrial producer of valuable organic acids, there are no comprehensive reviews of *M. elsdenii* in the literature. This review seeks to address this gap through the summative characterization of this species’ isolation, metabolism, growth behavior, abundance, and ecology as well as its effects on ruminant production and its potential application both as a probiotic for ruminants and as an industrial producer of carboxylic acids.

## 2. Classification and Morphology

*M. elsdenii* is a Gram-negative, strictly anaerobic, coccus-shaped, nonmotile bacterium that inhabits the rumen of cattle and other ruminants [7] as well as the large intestine of humans [9] and pigs [10]. It is classified within the phylum Firmicutes (syn. Bacillota) and class Clostridia [11], although it was previously classified in class Negativicutes. *Megasphaera* is one of a few genera within its phylum to have a porous pseudo-outer membrane that results in a negative Gram-staining reaction, although details of its lipopolysaccharide structure remain sketchy [12]. Within the the genus Megasphaera are included several species as follows: *M. elsdenii, M. hominis*, *M. cerevisiae*, *M. micronuciformis*, *M. paucivorans* and *M. sueciensis*, which have been isolated from various sources, including the rumen, human clinical specimens, and spoiled beer [13].

The name *Megasphaera* originated from the distinctive morphology of this bacterium, i.e., a large cell (mega = Latin for “large”) having a spherical shape (coccus) [14] (Figure 1). Huhtanen and Gall [7], who first described the bacterium now known as *M. elsdenii*, isolated it from calves or adult cattle fed high-grain diets, noted their relatively large size and termed them as a RO-C8 type LC (“large coccus”).

However, the name *Megasphaera* is a bit misleading. Although *M. elsdenii*’s cell size (2–2.5 μm diameter) exceeds that of the streptococci and staphylococci (0.7–1.5 μm) known at the time of its first description, it hardly compares to those of the largest coccoid bacteria discovered since then, such as *Thiovulum majus* (18 μm [15]) or the gigantic *Thiomargarita namibiensis* (750 μm [16]).

Huhtanen and Gall [7] grew *M. elsdenii* in vitro and observed that it fermented lactate and produced C_2_–C_6_ VFAs (acetic, propionic, butyric, valeric, and caproic) as well as CO_2_. Subsequently, Elsden et al. [17] isolated a bacterium (also termed the LC organism) from the rumen of sheep and presented similar descriptions of its morphology and physiology. Gutierrez et al. [18] also described a rumen isolate from bloated cattle fed high-grain diets as an “LC type” and proposed the name *Peptostreptococcus elsdenii*. Cells generally presented themselves in pairs and short chains (from 4 to 8 cells), which produced C_2_–C_5_ VFAs, CO_2_ and H_2_ when fermenting lactate. The authors proposed that *P. elsdenii* probably was important in stabilizing ruminal chemistry due to its capacity to ferment lactic acid. Later, Rogosa [14] proposed its reclassification from *Peptostreptococcus elsdenii* to *Megasphaera elsdenii* based on differences between the ruminal isolates [17,19] and the previously described members of the Gram-positive, actively proteolytic genus *Peptostreptococcus*, [20]. The LC-1 strain of Gutierrez represents the original lineage of the species’ currently recognized type strain, ATCC 25940 (=DSM 20460).

From an evolutionary perspective, Piknova et al. observed that strains of *M. elsdenii* isolated from lambs, calves, and sheep had highly similar 16S rRNA gene sequences [21]. These authors posited that the low genetic variability among *M. elsdenii* strains suggests a recent divergence from a common ancestor. Similarly, Shetty et al. [13] observed that two strains of *Megasphaera* isolated from the human gut showed high similarity in the 16S rRNA gene sequence between *M. hominis* and *M. elsdenii*. Figure 2 shows a rooted phylogenetic tree with all species of the genus Megasphaera in which all rumen-derived strains (*M. elsdenii*), which are clustered together, are compared with those from monogastric or beverage sources. The *Megaphaera* genus itself appears to be distinct and well separated from related genera (*Dialister, Veilonella*). Based on this analysis, the single strain of *Anaeroglobus* embedded within the *Megaphaera* genus would appear to be a likely candidate for reclassification.

At the time of writing, there are 53 *M. elsdenii* genomes in various states of assembly in the NCBI database of which only three have been completed [22,23,24]. One strain of the close relative *M. indica* has also been fully sequenced. (Table 1).

These genomes contain a single circular chromosome of ~2.2 Mbp, encode for ~2200 proteins whose coding regions average ~53 mol % G+C, and lack extrachromosomal elements. Their modest genome size is toward the lower end for bacterial genomes (range 0.6–14.3 Mbp) and is among the smallest for phylum Firmicutes, whose genome sizes span from 1.5 to 5.5 Mbp [25]. The relatively small genome of *M. elsdenii* is consistent with its modest catabolic and biosynthetic capacity and its strong symbiotic associations with its host and the host’s microbiota, all of which are factors that allow for the selection of a smaller genome size in the bacterial world.

The genes in the type strain identified by Marx et al. [22] encode proteins which can be grouped into subsystems, according to Shetty et al. [13], from which the more abundant ones are as follows: (1) amino acids and derivatives (273 proteins, 11% of total); (2) carbohydrate metabolism (201 proteins, 9%); (3) cofactors, vitamins, prosthetic groups, and pigments (141 proteins, 6.4%); (4) protein metabolism (136 proteins, 6%); (5) RNA metabolism (115 proteins, 5%); (6) cell wall and capsules (108 proteins, 4.9%); and (7) DNA metabolism (94 proteins, 4%). Such a distribution reflects its niche specialization. For example, its proportion of genes associated with amino acid metabolism are substantially higher than in the ruminal cellulolytic specialist *Fibrobacter succinogenes*, which neither imports significant amounts of amino acids nor catabolizes them [26].

Shetty et al. [13] reported that the genome of the ruminal isolate *M. elsdenii* DSM 20460 had low similarity to genomes of human gut isolates, where only 252 of the former’s proteins displayed more than 99% similarity to those from humans. Additionally, they also found that more than 400 proteins encoded by the genes from human gut isolates were not observed in the ruminal isolate. These include genes related to bile resistance, various sensory and regulatory systems, stress-response systems, membrane transporters, and resistance to antibiotics, as well as those in the presence of diverse and unique sets of carbohydrate-active enzymes (CAZymes) amongst these isolates, with a higher collection of CAZymes being identified in the human gut isolates that are not found in the rumen isolates’ genomes. The authors attribute these differences to differences in the host’s diet, which are suggestive of the host-specific adaptation undertaken by these isolates (i.e., a much more extensive fiber degradation in the rumen that is mostly carried out by specialist taxa known for their robust fiber degradation capacity).

## 3. Abundance in the Rumen

Considering that all the strains of *M. elsdenii* isolated from the rumen up to now prefer to ferment lactate rather than glucose or other sugars, and that lactate concentration in the rumen is higher in animals fed high-grain diets due to active starch degradation, the abundance of *M. elsdenii* in the rumen tends to be higher in starch-fed animals [27,28,29]. Because lactate is almost absent from the rumen of animals fed forage-based diets, *M. elsdenii* usually is either not found or is detected in much lower abundance in forage-fed ruminants.

Early studies quantifying the abundance of *M. elsdenii* in the rumen relied on plate counting using lactate agar media [18,27,30] (Table 2).

In certain studies, the authors did not measure *M. elsdenii* numbers, but they instead enumerated total lactate-fermenting bacteria, which the authors assumed mostly comprised *M. elsdenii* (a reasonable assumption at the time but now probably an overestimation due to the more expansive catalog of ruminal microbial diversity). Mackie and Gilchrist [30] observed that in animals fed forage-based diets (only 10% grain), the total number of lactate-fermenting bacteria represented about 0.2% of the total cultivable bacteria, while in animals fed high-grain diets (71% grain) lactate-fermenting bacteria represented 22.3% of the total cultivable ruminal bacteria, which supports the association between lactate producers and lactate fermenters. The involvement of *M. elsdenii* in ruminal lactate fermentation is supported by the observation of a positive relationship between the relative abundances of *M. elsdenii* and Gram-positive rods (probably lactobacilli) in eight-week-old calves [27] [Hobson et al., 1958]. Similarly, Yang et al. [31] found that in biofilms formed in the grains of barley and corn incubated in the rumen of dairy cows, the lactobacilli were dominant members associated with starch digestion, which were followed by an increased abundance of the genus Megasphaera.

Since the development of culture-independent molecular methods, quantification has been achieved using a qPCR based on the 16S rRNA gene and the massive parallel sequencing of 16S rRNA genes in community DNA. While early qPCR quantifications were expressed as cell densities (cells mL^−1^) [32,33], virtually all recent studies have expressed abundance as “relative abundance” (RA), i.e., as a percentage of the total bacterial 16 rRNA gene copies. Such quantifications can be misleading due to the wide variation in copy number of these genes across taxa. All three completed *M. elsdenii* genomes contain seven copies of the 16S rRNA gene. Because ~80% of sequenced bacterial genomes have fewer than seven copies of this gene [34], using 16S rRNA gene copy numbers in community analysis will likely overestimate *M. elsdenii* abundance. The extent of this overestimation is unclear because the average copy number of the bacterial community in the rumen is unknown. Nevertheless, 16S-based quantification can be useful for the comparison of abundances across animals and treatments, particularly within individual studies.

Table 3 presents *M. elsdenii* abundance in the rumen of animals determined through these culture-independent methods [35,36,37,38,39,40,41,42,43]. Relative abundance (RA) in the rumen is typically well under 1% of the total bacteria, except in animals experiencing metabolic or production problems, particularly milk fat depression. However, the single study of the ruminal bacterial transcriptome [43] yielded much higher RA than parallel 16S gene sequencing methods did, suggesting that *M. elsdenii* may be more metabolically active than its low 16S RNA-gene based RA values would suggest.

Several studies [44,45,46] have compared population sizes of *M. elsdenii* relative to a baseline or control treatment without an indication of the abundance relative to the total bacterial community. Fernando et al. [44] reported an 11-fold increase in *M. elsdenii* RA in beef steers within 21 d of switching from a hay diet to a concentrate diet. Khafipour et al. [45] reported that in Holstein cows, induction of mild or severe subacute ruminal acidosis (SARA) through grain feeding increased the RA of *M. elsdenii* 16- and ~1000-fold, respectively, while induction of mild SARA through alfalfa pellet feeding actually decreased its RA approximately 16-fold. However, McCann et al. [46] observed no significant change in *M. elsdenii* RA upon SARA induction. He et al. [47] reported a twofold increase in the relative abundance of *M. elsdenii* in non-lactating Saanen goats undergoing induction of SARA (elicited through high-concentrate feeding) compared with herdmates not subjected to SARA induction.

## 4. Isolation and Cultivation

Despite claims to the contrary [48], the enrichment and isolation of *M. esldenii* are rather straightforward. Considering that this species is a lactate fermenter, ruminal samples from animals fed grain-based diets are the recommended environmental source for enrichment. Use of lactate as an energy source for enrichment and isolation is recommended based on the bacterium’s preference for lactate as well as the fact that (unlike glucose) lactate is fermented by few species of ruminal bacteria. Media such as LTY (a lactate/trypticase/yeast extract medium) provides sources of energy, amino acids, and vitamins [49]. Because another ruminal bacterium, *Selenomonas ruminantium* var. *lactilytica*, can also ferment lactate, the selection of *M. elsdenii* is facilitated through addition of monensin to which the former species is much more sensitive. Because *M. elsdenii* produces H_2_ [27,49], the addition of NaMoO_4_ plus 1,8-dihydroxyanthraquinone and 0.01 mM of 2-bromoethanesulfonic acid plus chloroform will inhibit the growth of symbiotic hydrogenotrophic sulfate reducers and methanogens, respectively [50].

Unlike most ruminal bacteria, *M. elsdenii* does not require a highly negative environmental redox potential. As a result, it is not necessary to add chemical reducing agents such as cysteine to an anoxic culture medium as long as the requirements for sulfur are met through addition of trypticase and/or yeast extract; in their absence, small (sub-millimolar) amounts of cysteine or other sulfur sources may be required for maximum growth. However, larger additions of cysteine should be avoided as *M. elsdenii* can ferment cysteine into H_2_S, which combines with ferrous ions in the medium to produce black FeS precipitates that interfere with the visual detection or quantitative measurement of cell growth (Figure 3).

## 5. Nutrient Metabolism and Growth

*Growth requirements.* Although *M. elsdenii*’s unique ability to use either lactate or glucose as a fermentable energy source was discovered upon its first isolation [7], a more detailed characterization of its growth requirements developed more slowly. Bryant and Robinson [51] observed that factors found in yeast extract, such as biotin, pantothenate, and pyridoxine, stimulated the growth of *M. elsdenii*, while acetate was required when glucose was used as energy source. Although Hungate [2] (p. 88) indicated that *M. elsdenii* required some amino acids for growth, Forsberg [52] observed that strains B159 and T81 could grow in a casein-free medium containing minerals, ammonium as the N source, a sulfur source, acetate, and glucose as a carbon and energy source along with the above-mentioned vitamins. This observation is consistent with that of Lee et al. [53], who reported the presence of all genes for the biosynthesis of 20 common protein amino acids. Subsequently, Wallace [54] observed that amino acid catabolism, while substantial, is of minor energetic significance. Rychlik et al. [55] observed that all the strains of *M. elsdenii* that they evaluated could grow on free amino acids but not on peptides, which suggests that *M. elsdenii* is deficient in peptidase activity. Unlike many ruminal bacteria, *M. elsdenii* does not require branched-chain volatile fatty acids (BCVFAs) as growth factors [2] (p. 88), thereby producing any as needed for branched-chain lipid biosynthesis through amino acid fermentation. Additionally, *M. elsdenii* tests negative for many classical diagnostic tests (production of indole or acetoin, gelatin liquefaction, nitrate reduction) [2] (p. 84).

Growth studies on chemostats have suggested that the growth yields of *M. elsdenii* B159 are atypically high (0.24–0.58 g cells [g glucose]^−1^) [56,57]. However, such values are substantially overestimated because they do not take into account the catabolism of the relatively high concentrations of amino acids in the complex growth media [58] or the non-catabolic conversion of glucose to intracellular glycogen [49]. Growth yields on a glycogen-free basis for strain T81 on glucose or lactate in media with low concentrations of amino acids were reportedly 0.079 and 0.024 g cells (g substrate)^−1^, respectively [49].

*Utilization of lactate and sugars.* The distinguishing metabolic feature of *M. elsdenii* is its ability to utilize both glucose and lactate as fermentable energy sources through well-established fermentation pathways.

Glucose is fermented through the Embden–Meyerhof pathway into pyruvate and thence into acetate, H_2_ and CO_2_ are produced through a classical phosphoroclastic pathway involving pyruvate dehydrogenase, with ATP produced through substrate-level phosphorylation. Acetate (C_2_) is further converted to butyrate (C_4_) and caproate (C_6_) through chain elongation (CE, also called reverse β-oxidation; Figure 4). Genomics-based metabolic modeling [53] (discussed further in Section 9 below) suggests that during the fermentation of glucose, crotonyl-CoA—a key intermediate in the CE process—is produced from pyruvate through a bifurcating pathway with AcCoA and succinate as alternative intermediates (see Section 9 below).

Lactate is subjected to dismutation, with an oxidative path to acetate (through pyruvate and generating ATP) and a reductive path to propionate (through an acrylyl CoA intermediate, the so-called acrylate pathway). Although the organism produces a NAD-independent D-lactate dehydrogenase (iD-LDH), an active lactate racemase (LR) assures that both D-lactate and L-lactate are utilized. Both acetate (C_2_) and propionate (C_3_) can be further converted through CE to produce valerate (C_5_) [8].

The recent whole genome completed for *M. elsdenii* NCIMB 702410 [23] supports the known capacity of this bacterium to ferment both substrates. Glucose fermentation is enabled by the genes for all the 10 enzymes involved in the EMP pathway as well as genes coding the enzymes necessary for lactate metabolism, including lactate racemase (four genes), lactate permease (two genes), and the lactate utilization protein (one gene). Contrarily, genes coding enzymes, associated with the metabolism of lactose, mannose, xylose, trehalose, cellobiose, and sucrose, were not found in the genome of *M. elsdenii*, which agrees with data presented by Marounek et al. [59] who observed that certain strains were unable to ferment or presented a weak reaction for those sugars. Moreover, genes encoding glucanases, hemicellulases, xylanases, and amylase activity were absent in the genome of *M. elsdenii* NCIMB 702410, a fact that is consistent with its known inability to degrade complex polymers from diets, such as cellulose, hemicellulose, and starch. Thus, the whole genome of *M. elsdenii* confirms its classification as a secondary fermenter in the rumen.

Much attention has been devoted to comparing *M. elsdenii*‘s behavior toward lactate with that toward sugars. Russell and Baldwin [60] reported that *M. elsdenii* B159 fermented glucose and maltose (sugars derived from the hydrolysis of starch) as well as sucrose and lactate, but it did not ferment xylose or cellobiose (sugars derived from structural carbohydrates). Although strain B159 could ferment glucose, maltose, and lactate simultaneously, lactate fermentation yielded slower growth (μ_max_ 0.21 h^−1^ versus 0.45 h^−1^ or 0.55 h^−1^ on glucose or maltose, respectively). Using chemostats, Russell and Baldwin [61] observed that strain B159 displayed higher affinity for glucose than for lactate (K_s_ = 0.111 vs. 0.37 mM), but this was not confirmed in later studies.

Marounek et al. [59] showed that lactate fermentation by *M. elsdenii* strains proceeded five to six times faster than glucose fermentation. Similarly, Hino et al. [62] observed that *M. elsdenii* NIAH102 fermented lactate six times faster than glucose, likely due to the fact that five to six times more lactate, on a molar basis, was needed to support a similar extent of growth as glucose.

Weimer and Moen [49] observed that complete consumption of lactate by *M. elsdenii* T81was observed at initial concentrations of up to 210 mM, while maximum glucose consumption was ~80 mM, and much of that glucose was stored intracellularly as glycogen rather than fermented. Moreover, when glucose and DL-lactate were available simultaneously, strain T81 used lactate first, and glucose utilization began only after lactate was almost completely consumed. Hino et al. [63] inferred that glucose is not fermented by *M. elsdenii* at higher rates until lactate racemase (LR) activity becomes minimal, which only happens when lactate is present at very low concentrations. These authors suggested that preference for lactate developed from its greater availability, and that intracellular LR activity seems to be the major means of controlling glucose fermentation. Preference for lactate is consistent with the constitutive synthesis of iD-LDH, priming the bacterium for immediate use when lactate is suddenly available during rapid fermentation of cereal grains.

*M. elsdenii*’s ability to ferment both glucose and lactate is likely of benefit in the sense that the feeding of highly digestible starch will yield both glucose (from starch depolymerization) and lactate (preferentially produced from glucose fermentation by streptococci or other community members). Moreover, *M. elsdenii* can accumulate large amounts of glycogen intracellularly for later use [49], which may provide a means of effectively competing against other community members for available glucose despite a lack of an innate capacity for high-flux glucose fermentation into lactate. Glycogen reserves also provide utilizable energy to support an unusually high maintenance requirement (*m* = 0.187 g of glucose [g cells·h]^−1^ [56]).

*End product formation.* The distribution of products of both glucose and lactate fermentation by *M. elsdenii* are summarized in Table 4. As noted above, the two substrates yield starkly different product profiles. Glucose fermentation yields VFAs with primarily even carbon numbers (acetate, butyrate, caproate), while lactate yields primarily acetate, propionate and valerate. In addition to VFAs, *M. elsdenii* also produces fermentation gases, including CO_2_ and a moderate amounts of H_2_ (0.27 mol [mol substrate]^−1^ by strain T81 on both glucose and lactate [49] and 0.036 mol H_2_ [mol lactate]^−1^ by strain LC-S [17]).

Reducing equivalents generated from sugar oxidation are disposed of through the reduction of protons to H_2_, or through the CE of acetate. CE can be further stimulated through the exogenous addition of acetate [63], which is stoichiometrically converted to butyrate with only modest additional caproate production [49]. Though H_2_ production diverts some reducing equivalents from the CE pathway, it appears necessary to provide substrate flux and entropic balance to the exergonic synthesis of VFAs in a manner similar to that used in the well-studied asaccharolytic ethanol-utilizing CE bacterium of *Clostridium kluyveri* [64].

Lactate fermentation proceeds through an acrylyl CoA intermediate, with the necessary reducing equivalents generated through the oxidation of a certain amount of lactate through the lactate dehydrogenase and the pyruvate dehydrogenase systems to produce acetate and ATP. A balanced dismutation for the simultaneous oxidation and reduction of lactate would yield the stoichiometry of 3 Lactate → 2 Propionate + 1 Acetate + 1 CO_2_. In practice, however, similar molar amounts of acetate and propionate are obtained, with the excess reducing equivalents produced from lactate oxidation from being used for the CE of a certain amount of propionate to valerate. Because acetate is also a potential substrate for CE, a certain amount of butyrate is also produced. Lactate racemase (LR) is a key enzyme in lactate metabolism performed by *M. elsdenii*. Not only does it allow the utilization of both the D-isomer and L-isomer of lactate (both of which are produced by the ruminal community) but it also serves as a key metabolic control point. Hino and Kuroda [65] observed that LR was not synthetized by *M. elsdenii* growing on glucose as an energy source, which explains why this bacterium does not produce propionate when fermenting glucose in the absence of lactate, which is the inducer of the LR [66]. Thus, it can be inferred that the contribution of *M. elsdenii* to propionate production in the rumen only will occur when sufficient lactate is available to stimulate LR synthesis. This property is very different from that in classical propionic acid bacteria, which produce propionate from glucose using the succinate pathway. Additionally, contrary to most other ruminal bacteria in which LDH is induced by the substrate, in *M. elsdenii* this enzyme is constitutive.

**Table 4 microorganisms-12-00219-t004:** Fermentation products of various strains of *M. elsdenii* grown in a batch culture on D-glucose or DL-lactate.

Strain	Time (h)	From Glucose							From Lactate							Reference
		S_i_ ^d^ (mM)	Mol % Product ^e^						S_i_ ^d^ (mM)	Mol % Product ^e^						
			For (C_1_)	Ace (C_2_)	Pro (C_3_)	But (C_4_)	Val (C_5_)	Cap (C_6_)		For (C_1_)	Ace (C_2_)	Pro (C_3_)	But (C_4_)	Val (C_5_)	Cap (C_6_)	
LC	nr ^a^								nr		26.7	22.4	22.4	28.6		[18]
LC-3	168	27.8	4.0	7.2	1.8	14.3	11.1	61.6	62	0	18.7	32.0	21.6	27.7	0	[17]
LC-S	nr ^a^								12.9		22.1	26.3	22.1	27.9	1.6	[17]
nr ^a^	nr ^a^	55.5	+	33.9	1.3	52.1	6.7	5.9	111		35.5	38.2	13.7	12.6	-	[27]
B159 ^b^	72	27.8		20.4	-	77.5	2.0	+	17.7		31.3	37.5	12.5	18.8	-	[52]
L8	20	10	27.9	-	-	48.6	2.7	20.7	20	-	29.7	30.6	18.0	21.6	-	[59]
LC-1			4.2	-	-	59.4	19.8	16.7		-	51.3	38.0	6.8	3.8	-	
J1			19.4	-	-	37.4	21.6	21.6		-	24.3	27.4	39.4	-	8.8	
AW106			0.7	29.2	-	47.4	11.7	10.9		9.9	40.1	29.2	7.0	4.7	-	
NIAH102	10	11.1		32.0	-	61.5	0.6	6.0								[63]
T81 ^c^	nr								35		43.1	50.4	3.4			[66]
T81 ^c^	48	41	-	16.4	-	45.5	0.7	37.3	100		36.8	37.4	13.1	10.7	0.1	[49]

^a^ nr, not reported. ^b^ = ATCC 17752. ^c^ = ATCC 17753. ^d^ Initial substrate concentration. ^e^ Molar amount of each VFA as a percentage of the total moles of the product. Most studies do not report the amount of substrate consumed, which is necessary for the calculation of the product yield per unit of substrate consumed. For, formate; Ace, acetate; Pro, propionate; But, butyrate; Val, valerate; Cap, caproate.

Fermentation end product values vary widely among strains (Table 4), though certain differences may be due to differences in growth conditions, particularly substrate concentration and incubation time. Higher substrate concentrations and longer incubation times promote a more “complete” fermentation through VFA chain elongation, enhancing the relative proportions of valerate and caproate.

The pH of the growth medium can also influence the end products generated by *M. elsdenii*. Counotte et al. [67] observed that the fermentation of DL-lactate into propionate through the acrylate pathway decreased with a decreasing ruminal pH, with a resultant increase in butyrate production due to the CE of acetate.

## 6. Resistance to Environmental Stress

*pH.* The pH is one of the most important characteristics of the chemical environment that affects living cells because it affects cell membranes and their functionality [68] along with intracellular properties, such as enzyme activities [57] and cell viability [69], as well as the thermodynamics and kinetics of many catabolic reactions [70]. Most ruminal microbes can be classified as neutrophilic organisms, which are defined as those that thrive at a pH of 6.5 to 7.5 [71]. Many ruminal microbial species, especially fibrolytic organisms, are sensitive to a low ruminal pH (<6.0), but certain species such as *Streptococcus bovis* and *Lactobacillus spp*.—both of which produce lactate—are resistant to a low pH [72].

Ruminants evolved to consume fibrous feedstuffs, which are slowly digested by ruminal microorganisms and stimulate the secretion of well-buffered saliva, thus keeping the rumen’s pH close to neutral (the pH range is from 6.2 to 7.0 in forage-fed animals [1] (p. 246), [72] (p. 9), and from 5.8 to 6.5 in well-adapted grain-fed cattle [73]). However, the increased demand for animal protein (meat and milk) by humankind in the last decades has pushed for improved animal performance, which can most easily be achieved by increasing the grain content of the animals’ diets. However, this strategy can decrease the ruminal pH to below 6.0 for many hours per day because the starch in grains is rapidly fermented (primarily into lactate) in the rumen. This can have negative effects on ruminal microbial populations and their activities and can lead to digestive disturbances, such as ruminal acidosis and its associated diseases (bloat, rumenitis, liver abscess, laminitis, etc.) [74].

Russell et al. [75] suggested that at a low extracellular pH, bacteria would have insufficient energy to export protons through the cell membrane to establish a protonmotive force, a principal reason why many bacteria display poor growth when the pH is decreased. In this way, these authors observed that *M. elsdenii* B159 grew in a medium containing glucose as an energy source only at pH > 6.1. Contrarily, Hobson et al. [27] observed that growth of the LC organism (*M. elsdenii*) occurred at pH values ranging from 4.8 to 8.1. Therion et al. [76] reported that *M. elsdenii* ATCC 25940 displayed more sensitivity to a low pH (5.0 to 5.5) when fermenting glucose than when fermenting lactate.

Weimer and Moen [49] verified that the growth rate (0.66 h^−1^) of *M. elsdenii* T81 on DL-lactate was not affected when the initial culture’s pH ranged from 5.0 to 6.6, but at pH 4.65 the growth rate was dramatically reduced (to 0.17 h^−1^). Rapid growth at pH 5.0 would help prevent lactate accumulation in the rumen even though this pH value is considered suboptimal for ruminal function. In contrast, Waldrip and Martin [77] noted that *M. elsdenii* B159 displayed similar rates of L-lactate uptake when growing at pH 6.0 to 8.0 and at higher rates at pH 4.0 to 5.0. The authors suggested that *M. elsdenii* probably takes up L-lactate through a proton motive force-driven mechanism.

*Antibiotics*. Antibiotics are used on farms to treat sick animals as well as to improve animal performance (i.e., as growth promoters), or they are used prophylactically to prevent digestive disturbances such as ruminal acidosis [78]. Thus, the ruminal microbial community is often exposed to antibiotics, and resistance mechanisms are critical for survival under these conditions. Antibiotic resistance can be an intrinsic characteristic of a microorganism (genes in the DNA) or acquired from other organisms through horizontal gene transfer via plasmids or transposons [79]. The resistance of *M. elsdenii* to common antibiotics used as growth promoters in ruminant diets, such as ionophores (e.g., monensin), is a critical aspect in its role in lactate fermentation in the rumen of feedlot cattle because other lactate fermenters in the rumen, like *S. ruminantium*, are known to be highly sensitive to these agents. Rychlik et al. [55] evaluated monensin resistance of certain *M. elsdenii* strains and concluded that monensin probably does not decrease *M. elsdenii* numbers in vivo at normal therapeutic doses.

Most of the common ruminal bacterial species are sensitive to tetracycline [80,81], but Flint et al. [82] isolated strains of *M. elsdenii* resistant to this antibiotic, two of which had a plasmid of approximately 5–8 kilobase pairs (kbp) that carried a tetracycline (Tc) resistance marker. Additionally, Nagaraja and Taylor [83] showed that *M. elsdenii* B159 was totally resistant to eight antimicrobial feed additives (lasalocid, monensin, narasin, salinomycin, avoparcin, thiopeptin, tylosin, and virginiamycin). The susceptibility and resistance of *M. elsdenii* to thirteen other antibiotics have also been reported by El Akkad and Hobson [80] and by Wang et al. [84].

*Organic acid toxicity. M. elsdenii* T81 cannot initiate growth at lactate concentrations of >210 mM at pH 6.8 [49] and is also sensitive to high extracellular VFA concentrations [85]. This inhibition is likely due in part to the well-known intracellular accumulation of VFAs in response to the passive migration of protonated VFAs into cells, which upon ionization at a higher intracellular pH result in high intracellular concentrations of VFA anions [86]. Other factors probably contribute to toxicity as VFA toxicity increases with chain length despite similar pK_a_ values for these VFAs. Other organic acids can also be inhibitory. For example, Prabhu et al. [66] have reported that the growth rate of *M. elsdenii* T81 (=ATCC 17753) was reduced fourfold by 30 mM of acrylate, a normally intracellular intermediate in lactate fermentation.

## 7. Relationships with Others Ruminal Microorganisms

The rumen is a complex habitat in which factors related to diets and to the host are major determinants of the resident microbial community. Interactions among community members are also important as modulators of ruminal ecology [87,88]. These interactions—mutualism, competition, predation and amensalism—all contribute to the rumen’s microbial community structure. Considering that *M. elsdenii* is not a primary fermenter in the rumen as it cannot degrade polymeric carbohydrates from diets (including structural or nonstructural carbohydrates), this bacterium depends on primary degraders that supply glucose or lactate which *M. elsdenii* can use as energy and a carbon source. Thus, in order to better understand the role of *M. elsdenii* in the rumen, it is important to consider its relationship with others ruminal microorganisms.

*Interactions with starch and sugar fermenters.* Assuming that *M. elsdenii* is probably the major lactate fermenter in the rumen and that rumen lactate primarily originated from *Streptococcus bovis* growing with high glucose concentrations, Russell et al. [89] examined the interaction between these two species in a continuous culture as a function of the pH and dilution (growth) rates. A binary coculture of the two species always displayed higher populations of *S. bovis* than those of *M. elsdenii*, which were attributed to the former’s higher growth rates, higher affinity for maltose, and lower maintenance coefficient. Increasing dilution rate (from 0.12 to 0.36 h^−1^ at pH 6.6) caused a dramatic increase in the *S. bovis*/*M. elsdenii* ratio (from 3.5 to 23). Lactate production by *S. bovis* was modest at pH 6.6 and 5.7 and at lower dilution rates (0.12 and 0.22 h^−1^) but became significant at pH 5.7 and high dilution rates (0.36 h^−1^). In this last condition (high lactate concentration) a lower *S. bovis*/*M. elsdenii* ratio was observed, which can be explained through the enhanced growth of the lactate fermenter *M. elsdenii* produced by *S. bovis*. However, at pH 5.4 the *S. bovis*/*M. elsdenii* ratio increased until no *M. elsdenii* cells were detected, indicating the sensitivity of this bacterium to a very low pH and a higher resistance of *S. bovis* to acidic conditions.

With the goal of examining the potential impact of *M. elsdenii* in both acute ruminal acidosis and SARA, Chen et al. [90] performed more systematic studies on the interactions among *M. elsdenii, S. bovis, S. ruminatium, Lactobacillus fermentum,* and *Butyrivibrio fibrisolvens* in batch cultures that were fed three different levels of starch at both pH 5.5 and pH 6.5. Under these in vitro SARA-simulating conditions, lactate consumption was largely able to keep pace with its production, which was apparently due to the contributions of both *M. elsdenii* and *B. fibrisolvens*, with the latter species displaying more positive correlations with the production of the degradation products of propionate, butyrate, and formate. Monocultures of *M. elsdenii* continuously fed 15, 30, or 90 mM of lactate displayed increased growth rates with an increased lactate concentration except for the very poor growth when fed 90 mM of lactate and at pH 5.5. These results run counter to the widely held perception that *M. elsdenii* is a superior lactate fermenter under SARA conditions. Examination of gene expression under the different growth conditions led Chen et al. [90] to suggest that under the more extreme conditions, *M. elsdenii* loses much of its capacity to keep pace with lactate production under acute ruminal acidosis conditions, but that limiting the negative effects of acidosis may be attainable by controlling the ruminal pH to a range where *M. elsdenii* can effectively operate. This conclusion is in accord with the study of Kung and Hessian [91], who showed that the inoculation of *M. elsdenii* B195 into in vitro ruminal fermentations of rapidly fermentable carbohydrates substantially attenuated lactate accumulation and the decrease in the pH.

Yang et al. [31] examined biofilms formed on the grains of barley and corn incubated in nylon bags inserted into the rumen of dairy cows. Genus *Lactobacillus* was the dominant taxon associated with starch digestion after 12 h of incubation, which was followed by an increased abundance of the genus Megasphaera. These results improve our understating of the ruminal microbial community not only through establishment of a tight relationship between these two genera but also by demonstrating that *M. elsdenii* is not restricted to the liquid fraction of ruminal contents. Thus, studies on the *M. elsdenii* population in the rumen need to evaluate its growth in both the liquid and the solid fractions of the rumen’s contents.

*Competition with lactate utilizers. Selenomonas ruminantium* subsp. *lactilytica* can be regarded as the chiefly known competitor of *M. elsdenii* for lactate produced in the rumen. However, the two species differ markedly in their catabolic pathways and their regulation (Table 5). *Selenomonas ruminantium* subsp. *lactilytica* conducts the mixed acid fermentation of glucose into acetate and propionate at low growth rates, but it switches to a homolactic fermentation at high growth rates [56] and, upon glucose exhaustion, can ferment lactate into propionate alongside the ancillary production of acetate and CO_2_ [92]. Unlike *M. elsdenii*, *Selenomonas ruminantium* subsp. *lactilytica* uses the succinate pathway for propionate production. In addition, the genome of *S. ruminantium lactilytica* TAM6421 is substantially larger and more complex than that of *M. elsdenii* is [93] (Table 5).

The current consensus is that *M. elsdenii* is the major lactate user in the rumen of animals fed high-grain diets, while *Selenomonas ruminantium* subsp. *lactilytica* is a significant but secondary contributor. This notion is supported by the physiological characteristics of pure cultures, enzymatic data, and labeling studies that can distinguish between the different routes of propionate synthesis (those of succinate versus acrylate pathways).

Like the non-lactilytic subspecies *S. ruminantium* subsp. *ruminantium*, *Selenomonas ruminantium* subsp. *lactilytica* is an outstanding glucose fermenter. Russell and Baldwin [60] showed that the lactilytic strain HD4 was capable of rapid growth, a high catabolic growth yield, and a very low maintenance coefficient on glucose (Table 5).

Russell and Baldwin [60] also observed that this strain displayed the catabolite repression of lactate utilization when either glucose or sucrose was added to the growth medium. In contrast, *M. elsdenii* displays lower growth yields on glucose and a much higher maintenance coefficient; is not subject to catabolite depression of lactate utilization by glucose; and displays superior growth properties on lactate. Counotte et al. [67] observed that pure cultures of *M. elsdenii* fermented DL-lactate more rapidly than *Selenomonas ruminantium* subsp. *lactiltyica* did (5.49 × 10^−14^ mol cell^−1^ h^−1^ versus 2.42 × 10^−14^ mol cell^−1^ h^−1^). Moreover, Fan et al. [94] reported that, on lactate, *M.elsdenii* BE2-2083 displayed higher maximum specific growth rates than *Selenomonas ruminantium* subsp. *lactilytica* HD4 did at both pH 6.5 and pH 5.5 as well as shorter lag times before initiating growth.

In addition to the differences in pathway regulation noted above, the two species differ in enzymatic capabilities. *M. elsdenii* has a lactate racemase that allows it to use both the D-isomer and L-isomer of lactate. Because *Selenomonas ruminantium* subsp. *lactiltyica* lacks both an L-LDH and lactate racemase, its lactate fermentation is restricted to the D-isomer. Additionally, D-lactate uptake is greatly enhanced through several dicarboxylic acids (malate, fumarate and/or aspartate) that are the intermediates or metabolites of the succinate pathway. This suggests that lactate utilization is more closely linked to (i.e., more dependent on) glucose metabolism.

Labeling studies also support the dominance of *M. elsdenii*. Early work revealed that increasing the levels of degradable starch in the diet, which facilitates lactate production, increased the relative contribution of the acrylate pathway to propionate production from glucose [95,96]. Later, Counotte et al. [67] used ^14^C-labeled lactate to demonstrate that *M. elsdenii* accounted for 60–95% of lactate fermented in binary cocultures with *S. ruminantium* subsp. *lactilytica*. Moreover, labeling experiments with mixed ruminal inocula in vitro indicated that ~74% of the lactate was converted through the acrylate path (characteristic of *M. elsdenii*), with the remainder converted through the succinate path (characteristic of *S. ruminantium* subsp. *lactilytica*). These results are supported by the more recent work of Fan et al. [94] who used ^13^C-NMR to show that in batch-mode binary co-cultures with *S. ruminantium* subsp. *lactilytica*, *M. elsdenii* accounted for 82% and 75% of ^13^C-D-lactate catabolism at pH 6.5 and 5.5, respectively. These percentages likely change within the rumen over time. The ability of *S. ruminantium* subsp. *lactilytica* to ferment sugars in preference to D-lactate suggests that its propionate formation in the rumen is maximal shortly after starch feeding (when glucose is likely to be most abundant), while propionate production by *M. elsdenii* occurs later, after lactate has accumulated.

An additional twist in the relationship between *M. elsdenii* and *S. ruminantium* subsp. *lactilytica* is suggested by their response to lipopolysaccharides (LPS), which are outer cell membrane components widely produced by Gram-negative bacteria, including those in the rumen. Sarmikasoglu et al. [97] have reported that mixed LPS from the rumen reduced the growth rates of pure cultures of *S. ruminantium* subsp. *lactilytica* HD4 and *Streptococcus bovis* JB-1 on glucose (i.e., during lactate production) but did not affect growth of *M. elsdenii* T81 on lactate. Because strain HD4 is also lactilytic and strain T81 can grow on glucose, additional tests are needed to determine if these inhibition differences are inherent features of the bacteria (independent of substrate) or are substrate-dependent.

While the interactions described above appear to be a sound model for lactate consumption patterns in the rumen, it is wise to recall that the rumen contains many bacterial species whose characteristics and functions remain to be elucidated. In this regard, two lactate-utilizing candidate species that are phylogenetically related to *Butyrivibrio fibrisolvens* and *Anaerococcus prevotii* have recently been identified from metagenomic analysis of ruminal lactate enrichment cultures [98]. The isolation, characterization, and quantification of these species in ruminal contents—along with the elucidation of their interactions with conventional lactate utilizers such as *M. elsdenii*—will ultimately improve our understanding of ruminal lactate metabolism.

*Interactions with other ruminal microbes.* In addition to its role in lactate fermentation, *M. elsdenii* also contributes to the rumen’s ecosystem by supplying ammonia and BCVFAs required by many species of ruminal bacteria [2] (p. 88) due to its active amino acid fermentation [54,99].

## 8. Nutritional Importance

The nutritional importance of *M. elsdenii* can be evaluated from four different perspectives on ruminant production as follows: (1) as a contributor to improved ruminant performance; (2) as a causative agent of milk fat depression in high-producing dairy cows; (3) as a probiotic to stimulate ruminal development in pre-weaning animals; and (4) as a probiotic to prevent ruminal acidosis in animals fed high-grain diets.

*Milk production efficiency.* Shabat et al. [100] compared the ruminal bacterial communities of 146 Holstein cows divergently grouped according to high or low milk production efficiency (MPE) when fed the same standard dairy ration (30% forage/70% concentrate) to avoid the confounding effect of diet composition on MPE. They reported that in the rumen of cows that display high MPE, there was an increase in the relative abundance of the genus Megasphaera (based on the 16S rRNA gene copy number measured through Illumina sequencing) as well as an increase in the expression of the genes associated with the acrylate pathway known to be characteristic of *M. elsdenii* lactate fermentation. On this basis, the authors ascribed a role for *M. elsdenii* in the elevated propionate levels observed in the rumen of these higher efficiency cows. However, *M. elsdenii* abundance in this study was very low (<0.08% of the 16S rRNA gene copy number) and thus would seem unlikely to have contributed substantially to the production of propionate, which is produced by many ruminal bacteria, particularly members of the abundant genus *Prevotella* [101]. Moreover, several other studies that compared cows of divergent efficiencies did not reveal differences in *M. elsdenii* abundance [102,103]. Yet, it is worth recalling that relative abundance does not equate to relative activity (as indicated by the transcriptome study of Park et al. [43]). The high maintenance requirements of *M. elsdenii* (i.e., high catabolism without growth) may translate into an outsized fermentation and VFA production activity that may partially explain its contribution to propionate in the Shabat et al. [100] study. The potential importance of *M. elsdenii* in improving MPE, while intriguing, remains uncertain and warrants further study.

*Milk fat depression.* Fat is one of the most important milk components and has been used for decades as way to determine milk price in various countries; however, it is also the component that has the highest variability. In Holsteins, the fat content in milk can drop below 3.0%, the commonly held standard for so-called milk fat depression (MFD). MFD is most common in high-producing dairy cows fed high-starch (low-fiber) diets, especially when supplemented with polyunsaturated fats and with the ionophore monensin. It is known that MFD is caused by the negative effect of certain *trans*-10 fatty acids—particularly *trans*-10, *cis*-12 conjugated linoleic acid (*t*10-*c*12-CLA)—on de novo fat synthesis in the mammary gland of which production is increased in the rumen of MFD cows [104]. The current challenge is to identify the major organisms that effectively produce these fatty acids in vivo.

Although *Butyrivibrio fibrisolvens* is the ruminal bacterium more frequently associated with lipid metabolism in the rumen [105], presenting potential to hydrolyze lipids and hydrogenate polyunsaturated fatty acids, this bacterium neither synthetizes *t*10-*c*12-CLA [106] nor can grow at a ruminal pH lower than 5.95 [57], conditions under which *trans-*10 fatty acid production in the rumen increases. In view of this, Kim et al. [107] isolated *M. elsdenii* YJ-4 from the rumen of a cow fed with 90% grains and observed in vitro that this strain, as well as *M. elsdenii* T81, produces considerably more *t*10-*c*12-CLA (>6 µg and >4 µg mg protein^−1^, respectively) than other strains evaluated (B159, AW106, and JL1). Later, Kim et al. [108] inferred that the production of *t*10-*c*12-CLA could be increased when ruminal fluid was enriched with lactate.

In accordance with these pure culture results, certain studies, such as those of [36,37], have shown that dairy cows displaying MFD presented higher relative abundance of *M. elsdenii* (up to 4% of the total bacterial 16S rRNA gene copy number compared with just 0.02% in cows before the challenge to MFD or in cows that did not present MFD). Based on these results, Weimer et al. [50] conducted three experiments in which *M. elsdenii* was dosed directly into the rumen of cannulated dairy cows. In some experiments, the dosed *M. elsdenii* strains were those that had been previously isolated from the same individual cow during an episode of MFD. The authors did not find any effect of dosing *M. elsdenii* in the rumen on milk production and composition or on the ruminal pH, total VFAs, and lactate in the rumen, although the butyrate proportion was increased after dosing. The authors also observed that in almost all animals the *M. elsdenii* abundance in the rumen dropped to low baseline levels (<0.02% of the 16S rRNA gene copy number) within 24 h of dosing even upon multiple dosing, which could explain the lack of the effect of dosing on ruminal parameters and, possibly, milk composition. At this point, the relationship between the ruminal abundance of *M. elsdenii* and MFD remains associative, but a causative role has not yet been demonstrated.

*Dosing to stimulate ruminal development.* Two studies have been published that aimed to stimulate ruminal development through inoculation of *M. elsdenii* into pre-weaned calves. Muya et al. [109] found that *M. elsdenii* NCIMB 41125 dosed at 14 days of age increased ruminal butyrate and plasma β-hydroxybutyrate (BHBA) and also improved both the intake of starter feed and ruminal development (measured as an increase in the absorptive area) compared with non-dosed calves. In the second study [41], Lactipro^®^ (2 × 10^8^ cfu/mL of *M. elsdenii* NCIMB 41125) was administered to dairy calves and, although *M. elsdenii* shifted the abundance of certain less abundant bacteria, there was no increase in *M. elsdenii* abundance as well as no effect on ruminal and blood parameters.

*Dosing to control lactic acidosis.* The ability of *M. elsdenii* to consume lactate produced from starch or sugars by *Streptococcus bovis* and other rapid fermenters has fostered research aimed at using *M. elsendii* as a probiotic to attenuate the effects of acute lactic acidosis (ALA), a serious metabolic condition that contributes to the disruption of the rumen’s microbial ecology, reduced ruminal fiber digestion, damage to the ruminal epithelia as well as the liver, and inflammation of the hoof wall (laminitis). Because ALA undesirably compromises both animal health and productivity, many of these studies have examined the effect of dosing on both. This research has yielded a wealth of sometimes contradictory data that appear to be highly dependent on experimental conditions. A recent meta-analysis by Susanto et al. included 32 published studies (27 of which were peer-reviewed) focusing on cattle and sheep [110]. These studies varied with respect to diet and route of addition (oral vs. ruminal cannula) and included a wide range of dosing levels (log_10_ cfu of 7.0 to 13.3); however, the frequency and timing of supplementation were not analyzed. The authors concluded that supplementation with *M. elsdenii* decreased ruminal lactate concentration and the proportion of acetate in VFAs while increasing ruminal pH, methane production, and the proportion of C_3_–C_5_ in VFAs. The same meta-analysis reported that *M. elsdenii* supplementation reduced the occurrence of health problems (scours, bloat, and liver abscesses) and improved some metrics of animal performance (average daily gain (ADG) and hot carcass weight). Interestingly, dry matter intake decreased with *M. elsdenii* supplementation, but the feed conversion ratio did not change, thereby raising the question of whether supplementation would compromise production. We discuss here some of the studies used in the meta-analysis in more detail to illustrate the range of responses observed. We have generally excluded published studies that were not peer-reviewed, even though some of these studies were conducted under commercial production conditions, which were sometimes carried out with impressive numbers of animals.

Robinson et al. [111] dosed a strain of *M. elsdenii* into feedlot steers and observed lower ruminal lactate concentrations, a higher ruminal pH, and a 24% greater DMI when compared with control animals. Contrarily, Klieve et al. [33] did not observe any differences for the ruminal pH and the lactate concentrations between steers inoculated with *M. elsdenii* and control steers, because *M. elsdenii* had also established itself in the rumen of animals from the control group upon the increased addition of grain to their diet. However, grain-fed cattle inoculated with *M. elsdenii* displayed a 100-fold increase in abundance in the first 4 days following inoculation and also demonstrated an established population of *M. elsdenii* 7–10 d earlier compared with uninoculated cattle.

Later, McDaniel [112], using feedlot steers fed a high-concentrate diet dosed intraruminally with 1.62×10^8^ CFU/mL of *M. elsdenii* strain NCIMB 41125, observed that inoculated steers had a higher ruminal pH and lower lactate concentrations than non-dosed steers. In the same way, Leeuw et al. [113] dosed feedlot steers with *M. elsdenii* NCIMB 41125 and found a 5.6% better ADG during weeks 3–5 of feedlot as well as lower morbidity for dosed steers compared with the non-dosed group. Later, Henning et al. [114] administered *M. elsdenii* to feedlot cattle prior to feeding them with high-concentrate diets and observed an increase in dry matter intake, ADG, and feed efficiency. Henning et al. [115] then evaluated nine strains of *M. elsdenii* in vivo and observed high variability among strains regarding the capacity to control ruminal pH, with some strains preventing lactate accumulation, and a drop in the ruminal pH during an acidosis challenge. Aikman et al. [116] administered *M. elsdenii* to lactating dairy cows and observed a decrease in the amount of time that the ruminal pH was <5.6 compared with cows that were not supplemented. However, Zebeli et al. [117] intraruminally inoculated mid-lactation dairy cows (fed a diet containing 45% concentrate) with 35 mL d^−1^ of *M. elsdenii* ATCC 25940 (MEGA) culture that contained 10^8^ cfu mL^−1^ of bacteria and observed only slight changes in plasma metabolites and milk composition.

Evaluating the persistence of orally administered *M. elsdenii* in the rumen of beef cattle fed a barley diet, Klieve et al. [118] noted that the populations of *M. elsdenii* of 10^3^–10^7^ cells mL^−1^ were observed in all inoculated steers after 3 d but were also detected in 70% of uninoculated steers (though at a density lower than 1 × 10^6^ cells mL^−1^). For both groups of steers, *M. elsdenii* populations rapidly increased by day 14 (densities of 10^7^–10^8^ cells mL^−1^) and remained stable for the remainder of the experiment. The authors noted that the rapid acquisition of *M. elsdenii* for both groups of steers could question the need to inoculate them with the bacterium.

Higher ADG and hot carcass weights were observed by Drouillard et al. [119] in steers dosed orally with 200 mL of *M. elsdenii* NCIMB 41125 (10^11^ cfu mL^−1^) than in steers of the control group. In same way, Ye and Eastridge [120] observed higher milk and fat yield in dairy cows (>3 lactations) dosed with 200 mL Lactipro^®^ (1 × 10^8^ cfu mL^−1^) compared with those of a control group.

Arik et al. [121] dosed *M. elsdenii* (ATCC 17753, 200 mL of 2.4 × 10^10^ cfu mL^−1^) in the rumen of cannulated Holstein heifers fed wheat or a corn-based concentrate diet and found that *M. elsdenii* inoculation helped to prevent SARA by decreasing *S. bovis* populations and increasing protozoal populations in the rumen. Sedighi and Alipour [122] used in vitro and in vivo studies to evaluate four strains of *M. elsdenii* and observed that *M. elsdenii* inoculation increased in vitro gas production and pH values as well as a decreased lactate concentration. In the in vivo study, the authors found that the oral dosing of *M. elsdenii* SA3 to dairy cows decreased ruminal lactate compared with those of the control and had a tendency to increase milk fat yield.

*M. elsdenii* NCIMB 44125 was also orally dosed (100 mL of 2 × 10^8^ cfu mL^−1^) into cull beef cows in two simultaneous studies in which the authors found an increase in the absorptive surface area (ASA) of the rumen wall and an increase in the papillae area to ASA ratio [123], as well as a tendency to increase the average daily gain and carcass weight, compared with those of the undosed control group [124]. These authors also used the same strain of *M. elsdenii* (50 mL, 2 × 10^8^ cfu mL^−1^), which was orally dosed in early weaned beef calves, and observed that dosed animals presented a higher ADG during the first 21 d of study as well as a tendency to have a higher percentage of intramuscular fat and higher marbling scores than undosed calves had. The authors observed a lower liver score for dosed animals, which suggests the benefits of *M. elsdenii* in preventing liver abscesses.

Mazon et al. [125] evaluated *M. elsdenii* NCIMB (100 mL, 2 × 10^8^ cfu mL^−1^) in lactating dairy cows dosed 4 d and 1 d before a SARA challenge. Dairy cows dosed 4 d before acidosis challenge presented a higher ruminal pH and experienced shorter and less intense acidosis and, as a consequence of this, had a higher dry matter intake and a higher milk yield than those of undosed cows.

More recently, Lopes et al. [126] observed that dosing 20 mL of Lactipro^®^ (*M. elsdenii*, 1 × 10^10^ cfu mL^−1^) into feedlot Nellore bulls allowed for the reduction of the length of the adaptation made to a feedlot diet by 6 d compared with that of the undosed control group.

From these many studies, it appears that the potential benefit of *M. elsdenii* in preventing ruminal acidosis depends on its dosage and probably on the capacity of the inoculated (dosed) strain to persist in the rumen after dosing, which in turn depends on its encountering of favorable conditions in the rumen (a permissive pH and sufficient lactate concentrations). Unfortunately, very few studies have quantified the dosed strains in the rumen over the time following inoculation. From a research standpoint, a holistic understanding of the effects *M. elsdenii* in controlling lactate-mediated ruminal acidosis will require an integration of the population dynamics of *M. elsdenii* and associated, interacting species with pH, metabolite (lactate and VFAs) concentrations, and animal performance metrics. From a producer standpoint, the development of stable commercial products for frequent oral dosing to animals is essential for establishing the strain in the rumen in commercial herds under production conditions, or at least for maintaining a sufficient abundance between administrations of dosing, in order to achieve the goal of controlling acidosis. Finally, it is worth mentioning that any success of the probiotic *M. elsdenii* in controlling acute lactic acidosis may be compromised somewhat by the conversion of lactic acid to other VFAs (particularly butyric) with a resulting increased propensity to trigger SARA [47].

The effectiveness of *M. elsdenii* in mitigating ruminal acidification might also be hypothesized to control certain foodborne pathogens such as *Escherichia coli* O157:H7, whose pathogenicity has been reported to be induced by ruminal acidosis prior to shedding from the hindgut [127]. However, oral drenching of finishing steers with *M. elsdenii* did not significantly affect the prevalence of *E. coli* O157:H7 in anorectal swabs, suggesting that they did not impact this particular metric of food safety [128].

## 9. VFA Production

*M. elsdenii* and the related ruminal isolates *M. hexanoica* [129] and *M. indica* [130] are among a select group of so-called chain-elongating (CE) bacteria that extend the length of short-chain fatty acids by successively adding acetyl units, thus converting acetate (C_2_) to butyrate (C_4_) and then to caproate (C_6_), or converting propionate (C_3_) to valerate (C_5_) (Figure 4). This CE occurs as part of energy metabolism (catabolism) and differs from the nearly universally distributed anabolic synthesis of long-chain fatty acids for cellular lipids in that, in CE, the carboxylate product is produced in large amounts, is exported from the cell, and does not proceed beyond the production of medium-chain carboxylates (MCC, ≤C8). Because MCCs can serve as feedstocks for producing alkanes, alkenes, or alcohols using chemical or electrochemical methods, there is substantial interest in this “carboxylate platform” for the fermentative production of these MCCs from sugars, from primary fermentation products, such as lactate or ethanol [8], or from complex feedstocks, such as food waste or cellulosic biomass [131]. These interests have spawned a variety of studies aimed at maximizing the production of these acids, and some of these have yielded impressive acid production values. Unfortunately, direct comparisons among these studies are complicated by wide variations in strain selection, reactor configurations, run times, and even analytical methodologies.

As noted above, *M.elsdenii* has the unusual property of producing even-chain fatty acids (C_2_, C_4_, C_6_) through the fermentation of sugars, but primarily produces odd-chain fatty acids (C_3_, C_5_) from lactate [49]. Because lactic acid is readily produced through the fermentation of sugars by various lactic acid bacteria (LAB), the conversion of low-quality sugar-containing wastes through a lactic acid intermediate can be carried out by LAB/*M. elsdenii* co-cultures, yielding valerate as the primary product at titers of 3.3 g/L [132], which is approximately twice that reported for pure cultures of *M. elsdenii* of non-ruminal origin on pure sugars [133]. Alternatively, *M. elsdenii* can produce caproate to high titers directly from the fermentation of glucose and other sugars. Roddick and Britz [134] have reported production of up to 4.3 g of butyric and 19 g of caproic acids per L by *M. elsdenii* ATCC 25940 in fermenters employing ion-exchange resins to adsorb VFA end products. Choi et al. [135] obtained the production of caproate by *M. elsdenii* NCIMB 702410 at titers of 28.42 g L^−1^ from sucrose, in a two-phase reactor. More recently, Nelson et al. [85] grew *M. elsdenii* NCIMB 702410 in fed-batch pertractive reactors to continuously extract VFA products and demonstrated the equivalent of 36.5 g of butyrate L^−1^ and 20.7 g of caproate L^−1^, among the highest titers reported for the microbial production of this acid mixture.

The CE process can be stimulated through the exogenous addition of VFAs, such as acetate, as electron acceptors if the amount of electron donors is in excess. As noted above, this phenomenon was first noted by Hino et al. [63] upon acetate additions to cultures of *M. elsdenii* NIAH102 growing on glucose. More recently, Jeon et al. [136] reported that the addition of C_2_–C_4_ VFAs to cultures of the related species *M. hexanoica* MH (Figure 2) enhanced the production of valeric, caproic, heptanoic, and octanoic acids (C_5_–C_8_), resulting in titers of 5.7, 9.7, 3.2, and 1.2 g L^−1^, respectively. Using lactate as the electron donor and acetate as the electron acceptor, *M. hexanoica* MH has been reported to produce titers of up to 3.7 g of caproate L^−1^ and 1.5 g of caprylate (C_8_) L^−1^ [137]. Additionally, only low concentrations of odd-chain acids (C_5_, C_7_) were produced despite the use of the odd-chain electron donor, lactate, further distinguishing this species from *M. elsdenii*. It should be noted here that the lack of the reported production of C_7_ or C_8_ by *M. elsdenii* may be largely due to experimental and analytical issues, including low concentrations of the electron donor, relatively short incubation times, and the use of HPLC methodologies that require very long retention times, with resultant difficulty in the detection of low C_7_ and C_8_ due to peak broadening.

The potential of *M. elsdenii* as a carboxylate producer has attracted the interest of metabolic modelers. Lee et al. [53] developed a model, dubbed iME375, from the reconstructed metabolic pathways encoded in the genome of strain DSM 20460, with particular attention paid to carboxylate biosynthesis. A bifurcated pathway for synthesis of the essential CE intermediate crotonyl-CoA from glucose was identified (Figure 5). Succinate is a key intermediate in the pathway and—unlike what has been observed in several ruminal bacterial species—its direction toward crotonyl-CoA is not diverted toward propionate due to the absence of genes encoding methylmalonyl-CoA mutase and methylmalonyl CoA decarboxylase [22,23]. Metabolic flux analysis indicated that the specific rate of caproate synthesis declined with an increasing growth rate due to decreased flux through the AcAcCoA arm of the bifurcated pathway. In contrast, there was nearly constant flux through the succinate arm of the pathway, which became dominant only when μ > 0.4 h^−1^. Caproate production could be stimulated through the exogenous supply of several pathway precursors, including acetate, succinate, or butyrate. The authors concluded that balancing the two arms of the pathway is key to maximizing caproate productivity, while simple gene knockouts would likely not improve productivity. Further genomic analysis suggested that CE reactions downstream of crotonyl-CoA display substantial redundancy in the form of separately encoded isoenzymes, which is not surprising given the important role of CE in *M. elsdenii* catabolism.

The iME375 represents a useful step forward in optimizing VFA production undertaken by *M. elsdenii*, but, as noted by Panikov [58], it has substantial limitations in that the underlying data did not take into account the substantial contribution of highly abundant supplementary nutrients (particularly that of amino acids and yeast extract) that are known to alter both fundamental growth parameters (Y_g_, *m*) and the flux of pathway intermediates. Regardless, there is general agreement [53,58] that metabolic modeling will play a major role in advancing our understanding of *M. elsdenii* metabolism and VFA productivity.

Two scenarios are readily envisioned for the use of *M. elsdenii* in industrial VFA production. First, this bacterium could be used either in pure cultures using lactate- or sugar-containing fermentation broths (producing mixtures of primarily odd- or even-chain VFAs, respectively) or in binary mixed cultures with *Streptococcus bovis* or other lactic acid producers using sugar-rich feedstocks (e.g., cheese whey or effluents of wet-fractionated herbage) to produce a spectrum of VFA products. Valerate, the major product of these latter fermentations (but not readily produced by most other CE cultures) has already been shown to be a platform chemical for the synthesis of automotive fuel [138]. However, the low titers of valerate in these fermentations (~3 g/L) would have to be substantially improved for economic feasibility. The second scenario would involve the incorporation of *M. elsdenii* into mixed culture reactor microbiomes that convert organic wastes to mixtures of medium-chain carboxylates (MCC, containing 5–8 C atoms). Such processes use complex naturallyderived “open-culture” microbial communities that allow for a wide range of feedstocks and operate without the need of sterilizing feedstocks or equipment [131]. These diverse bacterial communities can sustain considerable levels of MCC productivity, but generally produce low proportions of odd-chain MCCs [139], making them attractive targets for bioaugmentation using *M. elsdenii*. Ultimately, it should be possible to tune the fermentation to the desired mixture of VFA products whose subsequent chemical conversion would yield the desired type and proportions of fuel compounds (alkanes, alcohols, and ketones of the desired chain lengths). The ability of *M. elsdenii* to contribute to MCC production in open-culture reactor microbiomes was noted by Scarborough et al. [140] who reported that *M. elsdenii* represented between 46.3 and 14.2% of the bacterial community (based on 16S rRNA gene abundance) in anaerobic bioreactors fed thin stillage from ethanol fermentations and were maintained at pH 5.0 or 5.5, respectively. These bioreactors produced substantial yields of MCCs, and *M. elsdenii* abundance was highest during periods of maximum odd-chain VFA production, consistent with this species’ characteristic capabilities in this regard.

Finally, *M. elsdenii* may also be coupled with conventional anaerobic digestion to produce biogas. A recent example is provided by Luo et al. [141] who investigated the effect of the exogenous addition of *M. elsdenii* ATCC 25940 on the anaerobic fermentation of synthetic food waste in a leach bed reactor (LR) and on the subsequent biogas formation in a downstream up-flow anaerobic sludge blanket (UASB) reactor. Comprehensive mass balance analysis showed that *M. elsdenii* augmentation improved volatile solid removal in the LR (from 59 to 65%) while improving methanogenesis from both the LR off-gas and the VFA-rich leachate. These metrics were further improved through supplementation of the LR with acetate (carbon conversion efficiencies of 32, 38, and 43 per cent for the control, *M. elsdenii*-amended and *M. elsdenii*-amended with acetate, respectively). The improvements were ascribed to more efficient redirection of lactate from primary fermentation toward the CE of acetate to produce butyrate undertaken by *M. elsdenii* along with increased H_2_ production during CE.

## 10. Final Considerations

Despite its somewhat unusual metabolic properties and apparently outsized influence on its ruminant host, *Megasphaera elsdenii* remains relatively little studied. Our understanding of this bacterium would be greatly enhanced by focusing on several areas.

(1)Development of a facile genetic system, including the introduction and maintenance of stable extrachromosomal elements.(2)Characterization of population dynamics throughout the ruminant’s feeding cycle. In the case of studies using this species as a probiotic, this would include the measurement of population sizes during the entire “dosing cycle”, from the time of inoculation or feeding onward, as well as the identification of factors that limit persistence in the rumen.(3)Examination of more nuanced interactions with the host, such as sensing/production of metabolites or signaling molecules that facilitate interactions between *M. elsdenii* and both its host and other ruminal microbes.(4)Continued efforts to elucidate the relationship between *M. elsdenii* abundance and milk fat depression to identify a mechanism that would explain a relationship that is currently merely correlative.(5)Exploitation of the VFA chain-elongating capabilities of *M. elsdenii* either in pure cultures or in mixed communities containing other CE bacteria, with the goal of increasing the VFA yield and titer and of more effectively tuning the proportions of individual carboxylate products—particularly toward valerate, at the production of which *M. elsdenii* is very adept.

## Figures and Tables

**Figure 1 microorganisms-12-00219-f001:**
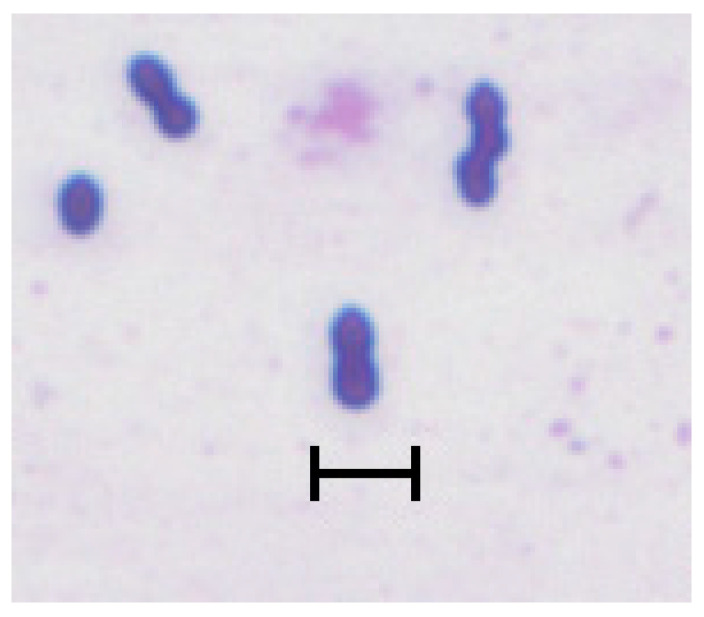
Photomicrograph of crystal violet-stained cells of *Megasphaera elsdenii* T81 grown in a lactate medium. Cells presented themselves singly, in pairs, or rarely as chains of 4–8 cells. The bar represents 5 μm. Photo provided through the courtesy of J. McClure, USDA–Agricultural Research Service.

**Figure 2 microorganisms-12-00219-f002:**
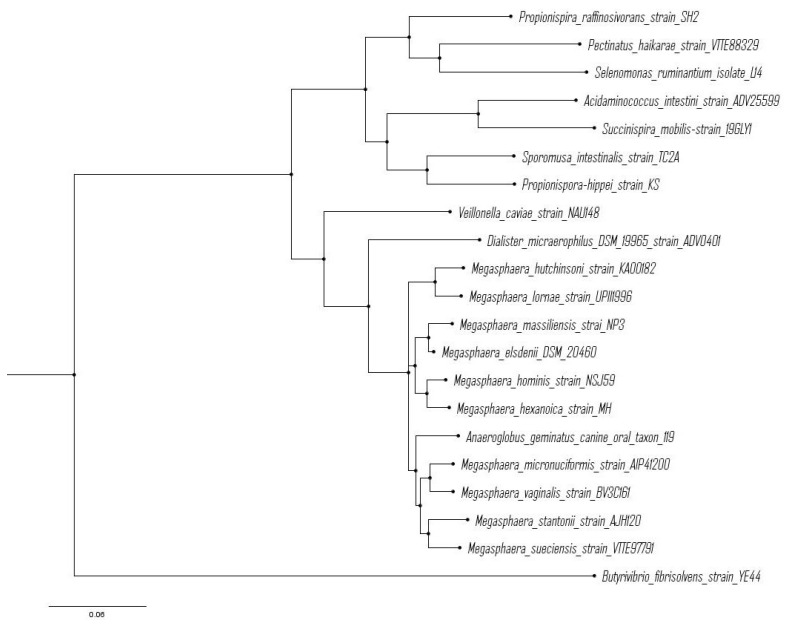
Unrooted phylogenetic tree for *M. elsdenii* and related species based on 16S rRNA gene sequences. The tree was constructed using mothur v.1.42.3 and Mr. Bayes 3.2.7a x86_64 software, with *Butyrivibrio fibrisolvens* as the outgroup, from sequences downloaded in FASTA format from the NCBI database. *M. elsdenii* strains ATCC 25940 and NCIMB 702410 were indistinguishable from the type strain DSM 20640^T^. Scale bar indicates nucleotide substitutions per site.

**Figure 3 microorganisms-12-00219-f003:**
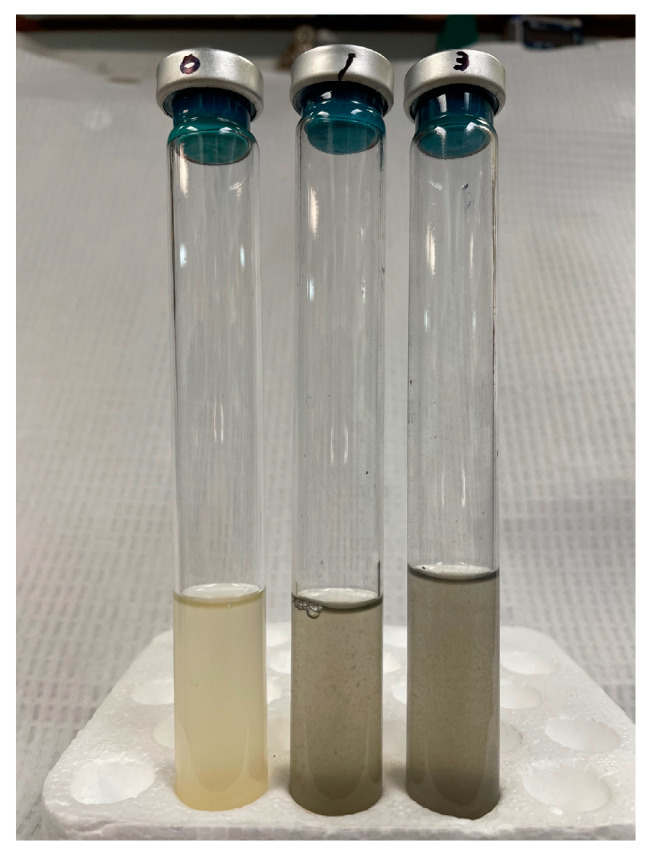
Cultures of *M. elsdenii* B159 grown in a LTY medium [49] containing (left to right) 0, 1, or 3 mM of added cysteine reducing agent. Blackening of the medium is due to an FeS precipitate formed following the fermentation of cysteine into H_2_S.

**Figure 4 microorganisms-12-00219-f004:**
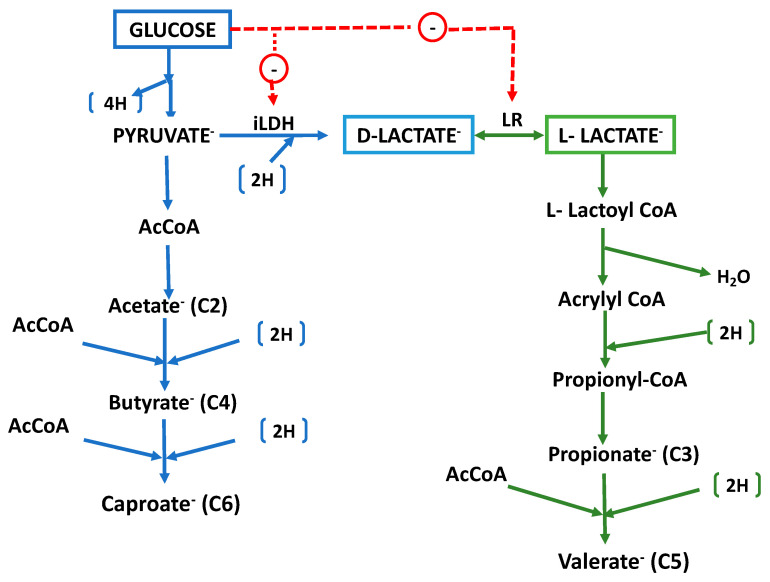
Simplified fermentation pathway for glucose and lactate by *Megasphaera elsdenii*, showing the key roles of acrylyl-CoA in the production of odd-chain fatty acids and acetyl-CoA (AcCoA) in the formation of both odd- and even-chain fatty acids. Glucose suppresses NAD-independent lactate dehydrogenase (iLDH) and lactate racemase (LR), preventing the significant production of odd-chain fatty acids from glucose.

**Figure 5 microorganisms-12-00219-f005:**
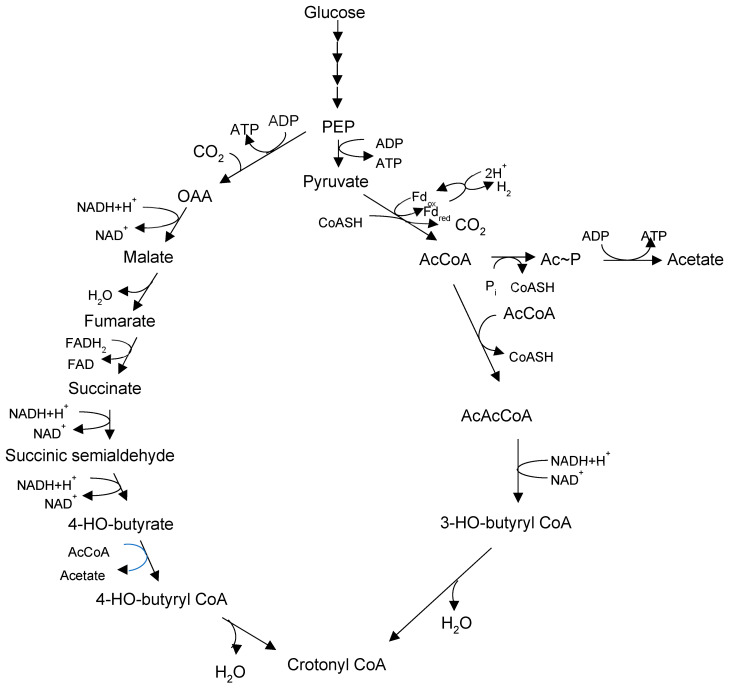
Bifurcated pathway for the biosynthesis of crotonyl-CoA, the key intermediate in chain-elongation of even-chain volatile fatty acids, by *M. elsdenii*. Crotonyl-CoA is reduced to butyryl-CoA for butyrate production or further elongated for caproate production. Adapted from Lee et al. [53].

**Table 1 microorganisms-12-00219-t001:** Sequenced genomes of *Megasphera elsdenii* strains and *M. indica*.

Strain	Source	Genome Size (bp)	Putative Number of Genes	Average G+C Content	Reference
DSM 20640^T a^	Rumen	2,474,718	2220	54	[22]
ATCC 25940 ^a^	Rumen	2,478,842	2194	52.8	[23]
NCIMB 702410	Rumen	2,566,193	2280	52.7	[23]
*M. indica*	Human feces	2,429,033	2184	53.2	[24]

^a^ These strains, though from different culture collections, are considered to have been derived from the same isolate.

**Table 2 microorganisms-12-00219-t002:** Total concentration of the *M. elsdenii* (LC) or lactate-fermenting bacteria (LFB) in cattle fed with different diets.

Organism Designation	Comparison	Counts/mL	Reference
Large coccus	Young calves (3–9 weeks) Adult cattle and sheep	10^9^ to 10^9^ absent to 10^5^	[27]
Large coccus	Before bloat Moderate bloat	<1 × 10^4^ to 2 × 10^7^ 2 × 10^6^ to 2 × 10^8^	[18]
Total lactate-fermenting bacteria	Forage diet High-grain diets	4.8 × 10^6^ 1.89 × 10^11^	[30]

**Table 3 microorganisms-12-00219-t003:** Relative abundance of *M. elsdenii* in the rumen.

Animal	n	% RA ^a^	Method ^b^	Notes	Reference
Holstein cows	2	0.0001–0.0011	qPCR	TMR, 27.5% NDF, 18% CP	[35]
Holstein cows	2	0.002–0.40	qPCR	High-concentrate diet. RA declined upon monensin feeding and withdrawal	[36]
Holstein cows	2 2	0.018–0.032 0.320–1.946	qPCR	Cows not fat-depressed Fat-depressed cows	[37]
Holstein cows	16	0.03–0.01	qPCR	30% forage/70% concentrate diet	[38]
Holstein calf to adult	5	0.0003–0.10	16S 454	Calf development study, diet varied over five time points from 1 d to 730 d age	[39]
Holstein cows	8	0.0001–0.0005 0.0025–0.0038	qPCR	Control or after recovery from MFD After induction of MFD	[40]
Holstein heifer calves	6	0 to 0.70	16S Illumina	Dosed with *M. elsdenii* probiotic at d14; RA declined to below uninoculated controls by d84	[41]
Crossbred steers	6	0.02 to 0.10	16S Illumina	6 diets varying in oilseed source, sampled 14 d after each dietary switch	[42]
Holstein bull calves (post-weaning)	5 5	0.379 0.790	RNA-seq	Inoculated with protozoal enrichment culture Uninoculated controls	[43]

^a^ Relative abundance, expressed as a percentage of the total 16S rRNA gene copy number. ^b^ Quantification method as follows: qPCR, quantitative polymerase chain reaction; 16S Illumina, 16S rRNA gene sequencing on the Illumina platform; 16S 454, 16S rRNA gene sequencing on the 454 pyrotag sequencing platform; RNA-seq, metatranscriptome analysis on the Illumina platform.

**Table 5 microorganisms-12-00219-t005:** Comparison of Megasphaera elsdenii and Selenomonas ruminantium lactilytica.

Property	*M. elsdenii*	*S. ruminantium lactilytica*
Phylum	Firmicutes	Firmicutes
Class	Clostridia	Negativicutes
Gram reaction	Negative	Positive
Motility	Nonmotile	Motile
Genome	Single chromosome ~2.5 Mbp	Single chromosome 3,003,680 bp, 9 plasmids 2619–285,449 bp. Total genome size 3,631,933 bp
Preferred growth substrate	D-lactate or L-lactate	D-lactate
Other growth substrates	Glucose, Fructose, Maltose, Amino acids	Glucose, Cellobiose, Mannose, Xylose, Arabinose, Mannitol, Glycerol
Lactate fermentation pathway	Acrylate	Succinate
Lactate utilization enzymes	NAD-independent L-lactate dehydrogenase, lactate racemase	NAD-independent D-lactate dehydrogenase. No L-LDH, no lactate racemase
VFA fermentation products ^a^	From lactate: C_2_, C_3_, C_4_, C_5_ From glucose: C_2_, C_4_, C_6_	From lactate: C_2_, C_3_ From glucose: lactate, acetate
Response to monensin	Relatively resistant	Sensitive
*m* ^b^ on glucose	0.187	0.021

^a^ C_2_, acetate; C_3_, propionate; C_4_, butyrate; C_5_, valerate; C_6_, caproate. ^b^ Maintenance coefficient (g glucose·g cells^−1^·h^−1^).

## Data Availability

No new data were created in this study. The 16S rRNA gene sequences used to construct the phylogenetic tree in Figure 2 were obtained from the National Center for Biotechnology Information, at https://www.ncbi.nlm.nih.gov/nuccore.

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
