# Peer review of "Megasphaera elsdenii: Its Role in Ruminant Nutrition and Its Potential Industrial Application for Organic Acid Biosynthesis"

_microorganisms, 2024, doi:10.3390/microorganisms12010219_

Round 1
Reviewer 1 Report
Comments and Suggestions for Authors
1. Insufficient Novelty: The article does not present new findings or does not significantly advance the current understanding of M. elsdenii beyond what is already known in the literature, it may lack the necessary novelty to warrant publication.
2. A review article should ideally be based on original research conducted by the authors. The article primarily summarizes existing literature without adding new experimental data or analytical insights, it may not meet the standards for originality.
3. The review is overly narrow in scope, focusing on a single aspect of M. elsdenii's biology or application without connecting it to broader contexts, it may not provide a comprehensive overview that would be valuable to the scientific community.
4. A review article should critically analyze the existing literature and the implications of the findings. If the article merely describes the research without critically evaluating its significance or limitations, it may not provide the depth of analysis expected in a review.
5. The article's conclusions are not well-supported by the data presented or do not offer clear implications for future research or applications, the article may not be seen as contributing significantly to the field.
In summary, the review article on Megasphaera elsdenii should be improved because it fails to meet the standards of originality, rigor, clarity, and relevance expected in scientific publications.
Author Response
Please see attached file, "RESPONSE TO REVIEWER 1 COMMENTS.pdf

Reviewer 2 Report
Comments and Suggestions for Authors
L113 chromsome/chromosome
L241 dcveloped/developed
L251 could grow on free amino acids but not on peptides
L506 “SARA conditions”?
L510 ·in vitro” sometimes in italics sometimes not
L613: Milk production efficiency: are the effects mentioned observed in cows fed diets with the same forage to concentrate ratio? More milk production may be associated with feeding concentrate rich diets, which in turn will favor M. elsdenii. Thus both a higher performance and a greater abundance of Megasphaera would be due to a concentrate-based diet, but not to a real direct relationship between those two variables. Please, discuss further.
L866 purê/pure
L911 why “facile”?
Author Response
We thank the reviewer for his/her comments. Our responses follow:
L113 chromsome/chromosome
AU: Corrected as requested (L121 of revised manuscript).
L241 dcveloped/developed
AU: Corrected as requested (L251).
L251 could grow on free amino acids but not on peptides
AU: Revised as suggested by the reviewer (L261-262).
L506 “SARA conditions”?
AU: The text “ARA” was correct as written. However, we agree that shifting the discussion from SARA to ARA (acute ruminal acidosis) is confusing, we so we have revised the text to spell out “acute ruminal acidosis” (L516-517).
L510 ·in vitro” sometimes in italics sometimes not
AU: As requested, corrected to italics in L507 and L520 for consistency.
L613: Milk production efficiency: are the effects mentioned observed in cows fed diets with the same forage to concentrate ratio? More milk production may be associated with feeding concentrate rich diets, which in turn will favor M. elsdenii. Thus both a higher performance and a greater abundance of Megasphaera would be due to a concentrate-based diet, but not to a real direct relationship between those two variables. Please, discuss further.
AU: The reviewer makes a good point, which we have now clarified. In the Shabat et al. study, the authors fed the both the high- and low-efficiency groups the same high production ration (30% forage/70% concentrate) to neutralize the effect of diet on efficiency. This is noted in the revised manuscript (L623-626).
L866 purê/pure
AU: Corrected as requested (L891).
L911 why “facile”?
AU: By “facile”, we mean easily used. Some genetic systems, particularly for anaerobic bacteria, are cumbersome and unlikely to be used even if they can technically be made to work. We have retained the word (L936).
Round 2
Reviewer 1 Report
Comments and Suggestions for Authors
We thanks for the authors' wonderful response and reply, and again it is ready for publication.